# Effects of *Acmella radicans* Invasion on Soil Seed Bank Community Characteristics in Different Habitats

**DOI:** 10.3390/plants13182644

**Published:** 2024-09-21

**Authors:** Xiaohan Wu, Kexin Yang, Fengping Zheng, Gaofeng Xu, Zewen Fan, David Roy Clements, Yunhai Yang, Shaosong Yang, Guimei Jin, Fudou Zhang, Shicai Shen

**Affiliations:** 1School of Agriculture, Yunnan University, Kunming 650504, China; wuxiaohan@mail.ynu.edu.cn (X.W.); yangkexin2202@163.com (K.Y.); 18539414083@163.com (Z.F.); 2Key Laboratory of Prevention and Control of Biological Invasions, Ministry of Agriculture and Rural Affairs of China, Agricultural Environment and Resource Research Institute, Yunnan Academy of Agricultural Sciences, Kunming 650205, China; zfping668@163.com (F.Z.); xugaofeng1059@163.com (G.X.); yangyunhainyx@163.com (Y.Y.); yshaos@163.com (S.Y.); jgmbly2006@126.com (G.J.); 3Key Laboratory of Green Prevention and Control of Agricultural Transboundary Pests of Yunnan Province, Agricultural Environment and Resource Research Institute, Yunnan Academy of Agricultural Sciences, Kunming 650205, China; 4Yunnan Lancang-Mekong Agricultural Bio-Security International Science and Technology Cooperation Joint Research Center, Kunming 650205, China; 5Department of Biology, Trinity Western University, Langley, BC V2Y 1Y1, Canada; clements@twu.ca

**Keywords:** *Acmella radicans*, soil seed bank, seed density, species diversity, invasive plant

## Abstract

To examine the effects of the recent *Acmella radicans* invasion on plant community and diversity in invaded habitats, the composition, density, species richness, diversity indices, and evenness index of the soil seed bank community of two different habitats (wasteland and cultivated land) in Yunnan Province, China, were analyzed through field sampling and greenhouse germination tests. A total of 28 species of plants belonging to 15 families and 28 genera, all annual herbs, were found in the soil seed bank. Seed densities and species number in the seed bank tended to be greater in April than in October; cultivated land also featured higher seed densities and species numbers compared to wasteland. With increased *A*. *radicans* cover, the seed bank population of *A*. *radicans* also significantly increased, but the seed bank populations of many other dominant species (e.g., *Ageratum conyzoides* and *Gamochaeta pensylvanica*) and native species (e.g., *Laggera crispata* and *Poa annua*) clearly declined. The germination of *A*. *radicans* seeds was concentrated during the period from the 4th to the 5th weeks. Vertically, the seed number of *A*. *radicans* was significantly different among the 0–5 cm, 5–10 cm and 10–20 cm layers that accounted for 80.7–90.6%, 9.4–16.1% and 0.0–3.2% of the total seed density in wasteland, respectively; and in cultivated land, *A*. *radicans* accounted for 56.8–64.9%, 26.7–31.8% and 8.1–13.5% of the total seed density, respectively. With reduced *A*. *radicans* cover, the species richness, Simpson index, Shannon–Wiener index, and Pielou indices of the weed community generally increased, and most diversity indices of weed communities in cultivated land were lower than in wasteland under the same cover of *A*. *radicans*. The results indicate that the invasion of *A*. *radicans* has negatively affected local weed community composition and reduced weed community diversity, and that these negative impacts in cultivated land may be enhanced by human disturbance. Our study was the first to elucidate the influence of *A*. *radicans* invasion on soil seed bank community characteristics in invaded habitats, providing a better understanding of its invasion and spread mechanisms in order to aid in developing a scientific basis for the prevention and control of this invader.

## 1. Introduction

The soil seed bank is usually considered as the natural reserve of all viable seeds in a soil profile and on the soil surface [1]. Many seeds may remain viable though dormant in the soil for years. According to persistence time in the soil, soil seed banks are categorized as transient seed banks (persist for less than 1 year), short persistent seed banks (persist for 1–5 years) and long-term persistent seed banks (persist for more than 5 years) [2]. As a potential source of genetic diversity for plant populations, soil seed banks are vital for vegetation succession, vegetation maintenance, ecosystem restoration, species diversity conservation, and population management strategies [3,4,5]. Therefore, it is of great value to study the species composition, and temporal and spatial dynamics, of soil seed banks to develop comprehensive weed management, sustainable agriculture, and conservation strategies.

Biological invasions have been become one of the main factors leading to biodiversity loss and environment change on a global scale [6,7,8]. The presence of seed banks has been regarded as one of the key components of successful biological invasion for many species [9]. The invasiveness of many exotic plant species is frequently associated with high proliferation and the production of seeds that may persist in the soil for long periods of time [10]. The successful establishment of invasive plants is determined in many cases by the production of a persistent soil seed bank in newly invaded areas, given the role of seed banks as sources of propagules, genetic diversity, and in spreading the risk of germination failure over time [9]. Therefore, studies on the soil seed bank of an invasive plant species can provide valuable information on the invasiveness of a species for planning restoration measures and for developing comprehensive prevention and control measures. This is especially important for invasive alien plant species that are new to a geographic area for which information on seed bank dynamics may be lacking.

*Acmella radicans* (Jacquin) R.K. Jansen is an annual erect herb in the Asteraceae family that originated in central America and Mexico [11]. This plant has become broadly distributed in Colombia, Bangladesh, Cuba, Curacao, India, Tanzania, and Thailand [11,12,13]. A naturalized population was first recorded in China in 2014 in Anhui Province [14]. In 2017, *A. radicans* was first found in Changning County, Baoshan City of Yunnan Province in southwestern China during a survey of invasive alien plant species. Currently, *A. radicans* has become a serious invasive species and has invaded farmland, tea gardens, orchard land, roadsides, and ditches in Baoshan City, Lincang City and Puer City of Yunnan [15]. *Acmella radicans* has become a dominant species in invaded regions, negatively affecting species richness, species diversity, and the evenness and soil nutrients of local communities. This plant can release allelochemicals such as (E, E)-2,4-decadienal, 2-tridecanone, and caryophyllene oxide to inhibit the seed germination and seedling growth of some major associated weeds and local vegetable crops such as *Ageratum conyzoides*, *Bidens pilosa*, *Brassica napus*, *Chloris virgata*, and *Digitaria sanguinalis* [15,16]. Furthermore, *A. radicans* demonstrates considerable ecological adaptability, occupies a wide range of habitat types, possesses a long flowering and fruiting period (November–March), and exhibits a prolific reproductive capacity (a single plant can produce up to 14,300 seeds) [15]. However, little is known about the soil seed bank in communities invaded by *A. radicans*.

In the current study, the composition, density, species richness, diversity indices, and evenness index of soil seed bank communities of *A. radicans* in two different habitats (wasteland and cultivated land) in Yunnan Province, China, were examined through field sampling and greenhouse tests. Seed bank characterization is important to better elucidate its invasion and spread mechanisms and provide a scientific basis for the forecasting, risk analysis, prevention, and control of *A*. *radicans*.

## 2. Results

### 2.1. Plant Species and Seed Density

A total of 28 plant species were identified within the study plots, belonging to 28 genera and 15 families (Table 1). Families with the most plant species were Asteraceae (9 species) and Poaceae (6 species), accounting for 32.1% and 21.4% of all species, respectively, and there was only one species for each of the other 13 families. All plants were herbaceous, among which 20 annual plants accounted for 71.4%, 7 perennial plants accounted for 25% and 1 annual/perennial plant constituted 4.6% of all species. In terms of invasiveness, there were 12 invasive alien plant species and 16 native species, accounting for 42.9% and 57.1% of all species, respectively (Table 1).

In terms of the frequency and density of species in the habitat, 12 plant species—*Acmella radicans*, *Ageratum conyzoides*, *Bidens pilosa*, *Chloris virgata*, *Digitaria sanguinalis*, *Eleusine indica*, *Gamochaeta pensylvanica*, *Galinsoga quadriradiata*, *Oplismenus undulatifolius*, *Poa annua*, *Polypogon fugax*, and *Spermacoce alata*—exhibited high population density and dominance (Table 2 and Table 3). As *Acmella radicans* cover increased, the seed density of *Acmella radicans* significantly increased; however, seed densities of many other dominant invasive plants, i.e., *Ageratum conyzoides*, *B. pilosa*, *E. indica*, *Gamochaeta pensylvanica*, *Galinsoga quadriradiata*, and native plants *C*. *virgata*, *D. sanguinalis*, *Laggera crispata*, *Oxalis corniculata*, and *P. annua* clearly declined. Compared with the total density of different habitats, the seed densities of invasive alien plants, native plants, and *A. radicans* accounted for 85.7–95.7%, 4.3–14.3% and 10.5–62.4% of the total seed density, respectively (Figure 1). With cover increases in *A. radicans*, the seed density proportion of invasive plants and *A. radicans* in both habitats observed was significantly increased, with the proportion significantly higher in cultivated land than in wasteland under the same *A. radicans* cover (Table 4).

### 2.2. Soil Seed Germination and Distribution of Acmella radicans

The germination tests showed that the germination process of *A. radicans* took place over 9 weeks, and that there was no difference in germination characteristics between wasteland and cultivated land. The seeds of *A. radicans* began to germinate after 2 weeks, with germination concentrated during the period from the 4th to the 5th weeks and ceasing at the 9th week.

The seed density and distribution of *A. radicans* were markedly different in different soil depths and habitats (Table 5). Seed densities of *A. radicans* showed obvious increases with increases in cover of *A. radicans* in both habitats, and the seed density of *A. radicans* in cultivated land was significantly higher than in wasteland under the same *A. radicans* cover. Vertically, the seed number of *A*. *radicans* was significantly different in shallower than deeper soil layers. The 0–5 cm, 5–10 cm and 10–20 cm layers accounted for 80.7–90.6%, 9.4–16.1% and 0.0–3.2% of the total seed density in wasteland, respectively; and in cultivated land, *A*. *radicans* comprised 56.8–64.9%, 26.7–31.8% and 8.1–13.5% of the total seed density, respectively (Figure 2). It is clear that *A. radicans* has persistent soil seed banks that play an important role during its invasion and expansion.

### 2.3. Effects of Acmella radicans on Soil Plant Species Diversity

Overall, the species richness (S), Simpson index (D), Shannon–Wiener index (H), and Pielou index (J) of weed communities were generally greater with less *A. radicans* cover, and most diversity indices of weed communities in cultivated land were lower than in wasteland under the same cover of *A. radicans* (Table 6). The results indicate that the invasion of *A. radicans* has negatively affected local weed community composition and reduced weed community diversity, and that these negative impacts in cultivated land may be enhanced by human disturbance.

## 3. Discussion

Increasing numbers of studies have demonstrated that invasive alien plants can reduce species richness, species diversity indices, and evenness indices through strong ecological adaptability, competitiveness and allelopathic inhibition, altering underground and aboveground composition and the structure of plant communities in invaded habitats [17,18,19,20]. As a new invasive species in Yunnan province, *A. radicans* is already widely distributed in Baoshan, Lincang and Puer cities, causing serious damage to local agricultural production and ecological environments [15]. However, the effect of *A. radicans* invasion on soil seed bank characteristics was previously unclear. Our current study found that *A. radicans* has modified the species composition, population density and weed community diversity of the soil seed bank in the invaded area, and human disturbance in cultivated land has aggravated these effects.

The soil seed bank is generally considered to be an important potential seed source for the invasion and spread of invasive alien plants and plays a critical role in vegetation succession and species diversity maintenance [17,18]. Characterizing the seed bank of invasive alien plants informs an understanding of the type, magnitude and rate of change in the seed bank and predicts the recruitment potential of these species from the seed bank [21]. The similarity between soil seed banks and the standing vegetation of invasive plants is usually high at the initial invasion stage and gradually declines over time [17,18]. Our current study found that the species richness of the soil seed bank is clearly higher than that of the aboveground vegetation, while the seed bank population is markedly smaller than the seed rain from the aboveground vegetation in *A. radicans* invaded habitats. A major factor contributing to this disparity is that a large proportion of the seed rain was transported to other places by wind and other disturbances. A second major factor is related to the invasion history and the storage characteristics of other plant seeds. As the invasion duration and resulting damage increases, invasive alien plants will gradually come to occupy a dominant position in both the local vegetation and the soil seed bank, while reducing the population density of other plants in the community, especially native species [22,23,24,25]. The current study found that as the seed density of *A. radicans* significantly increased, seed densities of many other dominant invasive plants and native plants clearly declined. The *A. radicans* seed density corresponded to increases in *A. radicans* cover, which was partly due to the strong competitiveness and allelopathic inhibition of *A. radicans* [15,16]. Moreover, with increased cover of *A. radicans*, the seed density of other invasive plants as well as *A. radicans* in study plots was markedly increased. The proportion of invasive species in the seed bank was significantly higher in cultivated land than in wasteland under the same *A. radicans* cover, indicating that *A. radicans* had higher negative impacts on native species than the other invasive alien species present, and that the weed community in cultivated land was affected by both the stress of *A. radicans* invasion and human disturbance. Local plant species and community structure are more vulnerable to disturbance than those of invasive alien plants under the same conditions because invasive species usually have greater morphological plasticity and competitiveness [26]. Synergistic impacts may result when adventive invasive alien plant species and disturbances coincide that are conducive to both the invasion of the adventive species and expansion of other invasive species [27].

Invasive alien plants not only change species composition and population density of weed communities in invaded habitats but also affect the species diversity. Zheng et al. [28] reported that the negative correlation between the importance value of *Spermacoce latifolia* and the species diversity of the community indicated that the invasion and spreading of *S. latifolia* may have a negative impact on the species diversity of the community. Nan et al. [18] found that the importance value of *Oenothera laciniata* in the community had a positive correlation with the species diversity of the community but this was not significant. Robertson and Hickman [29] found that as *Bothriochloa* spp. invasion increases, the native plant community decreased in diversity and abundance, but no effect on native seed bank diversity and density was observed. Our previous study found that *A. radicans* had negative impacts on species richness, species diversity, and evenness and soil nutrients of local communities [15]. Furthermore, we found that the species richness, Simpson index, Shannon–Wiener index, and Pielou index of the weed community usually increased as the *A. radicans* cover declined, and most diversity indices of weed communities in cultivated land were lower than in wasteland under the same cover of *A. radicans,* showing that the invasion of *A. radicans* has a negative impact on the diversity of local weed communities and that human disturbance in cultivated land may aggravate this trend. In our study area, there were few human activities (never grazed and/or cultivated) since 2002 in the wasteland habitat, indicating that the decline in species diversity of the weed community was directly caused by *A. radicans* invasion. The more pronounced decline in weed community species diversity in cultivated land was caused by the combination of *A. radicans* invasion and human disturbance. Other studies have reported that the invasion of exotic plant species and native species suppression were increased by numerous disturbances [30,31].

Invasive alien plants adapt to different environmental conditions and changes with various reproductive strategies during their life history [32]. Soil seed banks may be the most tolerant to a wide range of environmental conditions by comparison to other life history elements because many seeds may be dormant or remain viable in the soil over the long term [2,33]. The soil seed bank is heterogeneously distributed in soil wherein the topsoil layer is usually transient seed and has the highest proportion of the seed bank, while the proportion of the seed bank gradually decreases and becomes more of a persistent seed bank with increases in soil depth [34,35,36]. *Acmella radicans* is an annual herb, with a five-month flowering and fruiting period, producing up to 14,300 seeds per plant [15]. Its high fecundity is critical to the growth and spread of its populations. The present study showed that *A. radicans* seeds in the 0–5 cm, 5–10 cm and 10–20 cm layers of cultivated land accounted for 56.8–64.9%, 26.7–31.8% and 8.1–13.5% of the total seed density, respectively. In wasteland, most *A. radicans* seeds were stored in the 0–5 cm layer, accounting for 80.7–90.6% of the seed bank. The seed densities of *A. radicans* substantially increased with increases in the cover of *A. radicans* in both habitats, and the seed density of *A. radicans* in cultivated land was significantly higher than in wasteland under the same *A. radicans* cover. It is clear that *A. radicans* forms a large proportion of its persistent soil seed bank in deep soil in cultivated land, potentially contributing to future recruitment, especially if facilitated by human disturbance. Abundant transient and persistent seed banks of *A. radicans* are conducive to maintaining relatively high populations or even undergoing large outbreaks under the right conditions. Moreover, the persistent seed bank ensures *A. radicans* is not threatened with extirpation under adverse environment and human disturbance.

Considering the plant growth, reproduction, and soil seed bank characteristics of *A. radicans* under natural conditions, it is critical to adopt a long-term and comprehensive plan for its prevention, control and management. As an annual herb, *A. radicans* completes its life history and sexual reproduction within a single year. Thus, in order to reduce or avoid the seed dispersal of *A. radicans*, the optimum time for prevention and control is before flowering or at least prior to seed maturation with efficient herbicides or elimination by physical control techniques. In addition, for managing the transient and/or persistent soil seed bank *A. radicans*, a reduction in human activities or the use of soil herbicides to limit the seed germination after crop planting could be effective. Finally, prevention and control measures such as no-tillage mulching, plastic film mulching, intercropping, and replacement techniques could be combined, which would serve not only to inhibit the number of *A. radicans* effectively but also produce excellent ecological and economic benefits. Therefore, comprehensive measures must be considered for the prevention and control of *A. radicans*, incorporating chemical herbicides supplemented by physical control, ecological restoration, and replacement control. It is possible to control or reduce the extent of invasion by *A. radicans*, but this would require several years of continuous prevention and control measures.

## 4. Materials and Methods

### 4.1. Study Site

The study site was located in Changning County (20°14′ N–25°12′ N, 99°16′ E–100°12′ E), Baoshan City, in the west boundary area of Yunnan Province, southwest China. This area has a subtropical low-latitude mountain plateau monsoon climate, which is characterized by warm winter, cool summer, abundant rainfall, and wet and dry seasons. The annual average temperature of Changning is 14.9 °C, and the annual rainfall is 1259 mm. Recently, the range of *Acmella radicans* has been expanding rapidly within Changning County, as the plant has invaded cultivated land, orchard land, wastelands, riverbanks, and ditches [15].

### 4.2. Soil Collection

Two invaded habitats (wasteland and cultivated land) where *A. radicans* was growing were chosen in Mengtong Township, Changning County. The selected habitats were 5.5 km apart from each other and had similar soil types and ecological conditions but different levels of human disturbance. The wasteland is abandoned without any human activities, and the cultivated land is usually plowed with grown maize over one season and then it is fallowed for six months every year in these years. Plots 4 m × 4 m in size with less than 50 (35–50)% and more than 75 (75–100)% plant cover of *A. radicans* in each site were separately selected, and a total of 16 plots were randomly established and permanently marked.

The soil samples for analyzing the seed bank were collected in April and October 2022. *Acmella radicans* plants have completed fruiting, and most of them have fallen to the ground in April, and the natural germination of seed banks of this plant has finished before October. Five sample plots of 10 cm × 10 cm for each plot were randomly chosen at three soil layers (0–5 cm, 5–10 cm and 10–20 cm depths) for the soil collection. The soil samples of five plots at each layer were mixed to provide a composite sample. All the samples were properly marked and then transported to the laboratory. Litter and organic matter (including stones, coarse organic fragments, roots, and rhizomes) were removed from the soil samples; then, the samples were ground and sifted through a 4 mm sieve, which was followed by air-drying at room temperature for further germination tests. The soils from the two locations were similar and identified as yellow earths in two invaded habitats. The soils nutrient levels were measured as follows: pH 4.63–4.67, organic matter 30.96–31.41 g/kg, total N 1.49–1.53 g/kg, total K 1.04–1.21 g/kg, and total P 13.57–13.99 g/kg.

### 4.3. Germination Tests

Germination tests were conducted in the greenhouse to estimate the seed bank composition and density from 5 June to 13 October 2023. The soil was evenly distributed on flat plates (50 cm × 30 cm × 8 cm), and the soil sample depth was approximately 1–2 cm. The soils were regularly watered to avoiding drying out. After five weeks, the number of seedlings per plate was recorded, and seedlings were identified to species and removed each week. Those plants that could not be identified were transplanted and grown for later species identification. The germination test was terminated at the point when we no longer observed the emergence of new seedlings.

### 4.4. Statistical Analysis

Based on plant species identification and the seedling counting of soil samples, the species composition, seed density, seed proportion, and weed community diversity were analyzed. We combined the data from the three germination layers in order to estimate the seed density of each species in each soil collected times. Soil seed bank density (seedlings/m^2^) was calculated from the number of emerged seedlings in each plot with four replicates. The distribution percentage of invasive plants, native plants, and *A*. *radicans* was determined. Similarly, the soil seed bank density and distribution percentage *A*. *radicans* at each depth (0–5 cm, 5–10 m, 5–10 cm) and that of the whole profile (0–20 cm) were analyzed separately. Species richness, diversity and evenness were estimated as follows: (1) the Simpson diversity index (D) was calculated as D = 1 − ∑Ni (Ni − 1)/N (N − 1), where Ni is the total number of individuals from species i in a plot and N is the total number of individuals from all species in a plot; D ranges from 0 to 1, with 1 being the maximal diversity; (2) the Shannon–Wiener diversity index H [37] was measured as H = −∑Pi lnPi, where Pi is the proportion relative to the total number of species per plot; and (3) the Pielou evenness index (J) [38] was calculated as J = H/lnS, where S is the species richness of each plot. We tested the normality of the soil seed bank data and found the data were not normally distributed. Therefore, the nonparametric Kruskal–Wallis test was used for the identification of significant differences among groups at a 5% level using IBM SPSS22.0 software (Armonk, NY, USA).

## 5. Conclusions

Our results indicated that the invasion of *A. radicans* has changed the species composition, population density and weed community diversity of soil seed bank in the invaded area, and that human disturbance in cultivated land aggravated this trend. As *A. radicans* cover increased, the seed density of *A. radicans* significantly increased, but the seed densities of many other dominant invasive plants and native plants clearly declined. The soil seed germination of *A. radicans* occurred over 9 weeks and was concentrated during the period from the 4th to the 5th weeks. Seeds of *A. radicans* in the 0–5 cm, 5–10 cm and 10–20 cm soil layers of cultivated land accounted for 56.8–64.9%, 26.7–31.8% and 8.1–13.5% of the total seed density, respectively, but in wasteland, most seeds were stored in the 0–5 cm layer with *A. radicans* seeds accounting for 80.7–90.6% of the seed bank. The species richness, Simpson index, Shannon–Wiener index, and Pielou index of the weed community were generally increased with declining *A. radicans* cover, and most diversity indices of weed communities in cultivated land were lower than in wasteland under the same cover of *A. radicans*. The relationship between the soil seed bank and aboveground vegetation, influence of different environmental factors on seed germination, and effect of disturbance on the sexual reproduction characteristics of *A. radicans* require further research to help develop effective management strategies.

## Figures and Tables

**Figure 1 plants-13-02644-f001:**
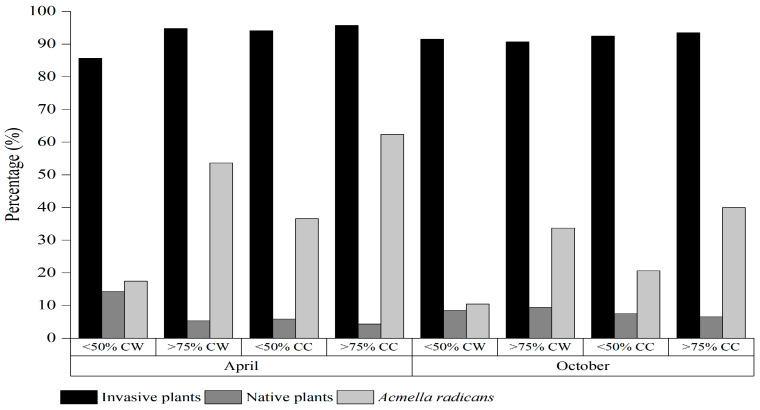
Percentage of invasive plants, native plants and *Acmella radicans* of the total density of soil weed communities within different habitats (CW = cover of wasteland and CC = cover of cultivated land).

**Figure 2 plants-13-02644-f002:**
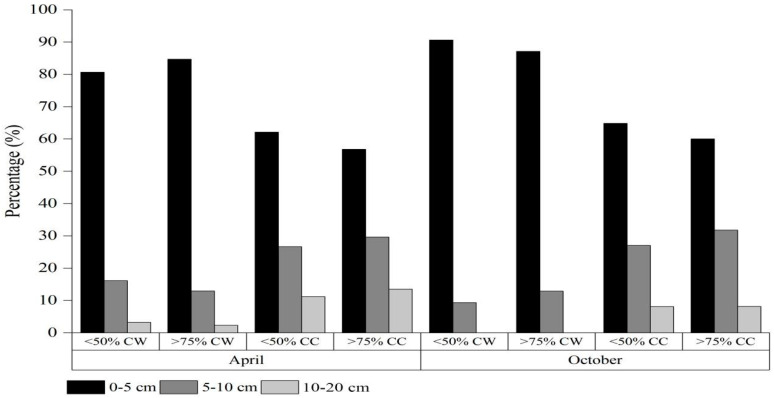
Percentage of soil seed distribution of *Acmella radicans* among different soil layers within different habitats (CW = cover of wasteland and CC = cover of cultivated land).

**Table 1 plants-13-02644-t001:** Basic information of plant species, life form and invasiveness of plant species found in the soil seed bank in study plots located in cultivated and wasteland habitats in Mengtong Township, Changning County.

Family	Scientific Name	Life Form	Origin
Amaranthaceae	*Chenopodium ficifolium* L.	Annual herb	China
Asteraceae	*Acmella radicans* (Jacquin) R.K. Jansen	Annual herb	Central America and Mexico
	*Ageratina adenophora* (Spreng.) R. M. King and H. Rob	Perennial herb	Mexico and Costa Rica
	*Ageratum conyzoides* L.	Annual herb	Tropical America
	*Bidens pilosa* L.	Annual herb	Tropical America
	*Crassocephalum crepidioides* (Benth.) S. Moore	Annual herb	Tropical Africa
	*Erigeron canadensis* L.	Annual herb	North America
	*Galinsoga quadriradiata* Ruiz et Pav.	Annual herb	Tropical America
	*Gamochaeta pensylvanica* (Willd.) Cabrera	Annual herb	North America
	*Laggera crispata* (Vahl) Hepper & J. R. I. Wood	Perennial herb	China
Brassicaceae	*Cardamine occulta* Hornem.	Annual herb	China
Caryophyllaceae	*Stellaria media* (L.) Vill.	Annual herb	Europe
Commelinaceae	*Commelina benghalensis* L.	Perennial herb	China
Cyperaceae	*Cyperus iria* L.	Annual herb	China
Linderniaceae	*Lindernia procumbens* (Krock.) Borbás	Annual herb	China
Onagraceae	*Oenothera rosea* L’Hér. ex Aiton.	Perennial herb	South and Tropical America
Oxalidaceae	*Oxalis corniculata* L.	Annual/Perennial herb	China
Phyllanthaceae	*Phyllanthus urinaria* L.	Annual herb	China
Poaceae	*Chloris virgata* Sw.	Annual herb	China
	*Digitaria sanguinalis* (L.) Scop.	Annual herb	China
	*Eleusine indica* (L.) Gaertn.	Annual herb	North America
	*Oplismenus undulatifolius* (Ard.) Roemer & Schuit.	Perennial herb	China
	*Poa annua* L.	Annual herb	China
	*Polypogon fugax* Nees ex Steud.	Annual herb	China
Polygonaceae	*Polygonum aviculare* L.	Annual herb	China
Ranunculaceae	*Ranunculus japonicus* Thunb.	Perennial herb	China
Rubiaceae	*Spermacoce alata* Aubl.	Perennial herb	South and Tropical America
Solanaceae	*Solanum nigrum* L.	Annual herb	China

**Table 2 plants-13-02644-t002:** Average seed densities of the soil seed bank within different habitats in April 2022 (seedlings/m^2^).

Scientific Name	Wasteland		Cultivated Land	
	<50% Cover	>75% Cover	<50% Cover	>75% Cover
*Acmella radicans* (Jacquin) R.K. Jansen	7750.0	22,325.0	18,275.0	38,775.0
*Ageratina adenophora* (Spreng.) R. M. King and H. Rob	50.0	0.0	0.0	0.0
*Ageratum conyzoides* L.	11,900.0	8575.0	16,650.0	12,100.0
*Bidens pilosa* L.	2350.0	925.0	1150.0	825.0
*Cardamine occulta* Hornem.	125.0	50.0	100.0	200.0
*Chenopodium ficifolium* L.	125.0	25.0	50.0	50.0
*Chloris virgata* Sw.	1175.0	525.0	225.0	0.0
*Commelina benghalensis* L.	150.0	25.0	175.0	275.0
*Crassocephalum crepidioides* (Benth.) S. Moore	250.0	25.0	75.0	50.0
*Cyperus iria* L.	450.0	50.0	375.0	225.0
*Digitaria sanguinalis* (L.) Scop.	725.0	400.0	75.0	75.0
*Eleusine indica* (L.) Gaertn.	1650.0	1100.0	1275.0	975.0
*Erigeron canadensis* L.	275.0	25.0	125.0	150.0
*Galinsoga quadriradiata* Ruiz et Pav.	775.0	300.0	625.0	500.0
*Gamochaeta pensylvanica* (Willd.) Cabrera	11,750.0	5750.0	7450.0	4925.0
*Laggera crispata* (Vahl) Hepper & J. R. I. Wood	300.0	50.0	100.0	50.0
*Lindernia procumbens* (Krock.) Borbás	100.0	175.0	100.0	75.0
*Oenothera rosea* L’Hér. ex Aiton.	100.0	0.0	0.0	0.0
*Oplismenus undulatifolius* (Ard.) Roemer & Schuit.	625.0	125.0	300.0	400.0
*Oxalis corniculata* L.	550.0	375.0	650.0	625.0
*Phyllanthus urinaria* L.	275.0	75.0	75.0	75.0
*Poa annua* L.	700.0	100.0	275.0	125.0
*Polypogon fugax* Nees ex Steud.	600.0	125.0	200.0	200.0
*Polygonum aviculare* L.	75.0	25.0	100.0	125.0
*Ranunculus japonicus* Thunb.	50.0	0.0	0.0	0.0
*Solanum nigrum* L.	275.0	75.0	150.0	200.0
*Spermacoce alata* Aubl.	675.0	400.0	1375.0	1125.0
*Stellaria media* (L.) Vill.	225.0	25.0	125.0	100.0

**Table 3 plants-13-02644-t003:** Average seed densities of the soil seed bank within different habitats in October 2022 (seedlings/m^2^).

Scientific Name	Wasteland		Cultivated Land	
	<50% Cover	>75% Cover	<50% Cover	>75% Cover
*Acmella radicans* (Jacquin) R.K. Jansen	800.0	1550.0	1850.0	2750.0
*Ageratina adenophora* (Spreng.) R. M. King and H. Rob	50.0	0.0	0.0	0.0
*Ageratum conyzoides* L.	2800.0	650.0	3425.0	1950.0
*Bidens pilosa* L.	0.0	0.0	0.0	0.0
*Cardamine occulta* Hornem.	25.0	0.0	50.0	25.0
*Chenopodium ficifolium* L.	0.0	25.0	0.0	0.0
*Chloris virgata* Sw.	75.0	100.0	0.0	0.0
*Commelina benghalensis* L.	0.0	0.0	0.0	0.0
*Crassocephalum crepidioides* (Benth.) S. Moore	50.0	75.0	50.0	100.0
*Cyperus iria* L.	75.0	25.0	50.0	50.0
*Digitaria sanguinalis* (L.) Scop.	25.0	50.0	25.0	75.0
*Eleusine indica* (L.) Gaertn.	400.0	300.0	250.0	200.0
*Erigeron canadensis* L.	50.0	0.0	100.0	50.0
*Galinsoga quadriradiata* Ruiz et Pav.	150.0	50.0	125.0	125.0
*Gamochaeta pensylvanica* (Willd.) Cabrera	2525.0	1575.0	2300.0	1150.0
*Laggera crispata* (Vahl) Hepper & J. R. I. Wood	25.0	0.0	50.0	0.0
*Lindernia procumbens* (Krock.) Borbás	100.0	75.0	100.0	75.0
*Oenothera rosea* L’Hér. ex Aiton.	50.0	75.0	0.0	0.0
*Oplismenus undulatifolius* (Ard.) Roemer & Schuit.	150.0	25.0	100.0	0.0
*Oxalis corniculata* L.	25.0	75.0	100.0	100.0
*Phyllanthus urinaria* L.	0.0	0.0	0.0	0.0
*Poa annua* L.	100.0	50.0	150.0	50.0
*Polypogon fugax* Nees ex Steud.	0.0	0.0	0.0	0.0
*Polygonum aviculare* L.	0.0	25.0	50.0	25.0
*Ranunculus japonicus* Thunb.	50.0	0.0	0.0	0.0
*Solanum nigrum* L.	0.0	0.0	0.0	50.0
*Spermacoce alata* Aubl.	75.0	75.0	175.0	125.0
*Stellaria media* (L.) Vill.	50.0	0.0	25.0	0.0

**Table 4 plants-13-02644-t004:** Kruskal–Wallis test results for the seed density of invasive plants, native plants and *Acmella radicans* in the soil seed bank within different habitats.

April			October		
Group	N	Mean rank of seed density in invasive plants	Group	N	Mean rank of seed density in invasive plants
<50% cover of wasteland	4	2.63	<50% cover of wasteland	4	10.00
>75% cover of wasteland	4	6.38	>75% cover of wasteland	4	2.50
<50% cover of cultivated land	4	10.5	<50% cover of cultivated land	4	14.50
>75% cover of cultivated land	4	14.5	>75% cover of cultivated land	4	7.00
Kruskal–Wallis test	13.988		Kruskal–Wallis test	13.500	
*p*	0.003	(Chi-square approximation)	*p*	0.004	(Chi-square approximation)
Group	N	Mean rank of seed density in native plants	Group	N	Mean rank of seed density in native plants
<50% cover of wasteland	4	14.50	<50% cover of wasteland	4	11.00
>75% cover of wasteland	4	4.25	>75% cover of wasteland	4	6.50
<50% cover of cultivated land	4	8.25	<50% cover of cultivated land	4	9.63
>75% cover of cultivated land	4	7.00	>75% cover of cultivated land	4	6.88
Kruskal–Wallis test	9.963		Kruskal–Wallis test	2.558	
*p*	0.019	(Chi-square approximation)	*p*	0.465	(Chi-square approximation)
Group	N	Mean rank of seed density in *Acmella radicans*	Group	N	Mean rank of seed density in *Acmella radicans*
<50% cover of wasteland	4	2.50	<50% cover of wasteland	4	2.50
>75% cover of wasteland	4	10.5	>75% cover of wasteland	4	6.50
<50% cover of cultivated land	4	6.50	<50% cover of cultivated land	4	10.50
>75% cover of cultivated land	4	14.50	>75% cover of cultivated land	4	14.50
Kruskal–Wallis test	14.159		Kruskal–Wallis test	14.264	
*p*	0.003	(Chi-square approximation)	*p*	0.003	(Chi-square approximation)

**Table 5 plants-13-02644-t005:** Kruskal–Wallis test results for the seed density of *Acmella radicans* among different soil layers within different habitats.

April			October		
Group	N	Mean rank of seed density in 0–5 cm layer	Group	N	Mean rank of seed density in 0–5 cm layer
<50% cover of wasteland	4	2.50	<50% cover of wasteland	4	2.50
>75% cover of wasteland	4	10.50	>75% cover of wasteland	4	9.50
<50% cover of cultivated land	4	6.50	<50% cover of cultivated land	4	7.50
>75% cover of cultivated land	4	14.50	>75% cover of cultivated land	4	14.50
Kruskal–Wallis test	14.118		Kruskal–Wallis test	13.234	
*p*	0.003	(Chi-square approximation)	*p*	0.004	(Chi-square approximation)
Group	N	Mean rank of seed density in 5–10 cm layer	Group	N	Mean rank of seed density in native plants
<50% cover of wasteland	4	2.50	<50% cover of wasteland	4	3.13
>75% cover of wasteland	4	6.50	>75% cover of wasteland	4	5.88
<50% cover of cultivated land	4	10.50	<50% cover of cultivated land	4	10.50
>75% cover of cultivated land	4	14.50	>75% cover of cultivated land	4	14.50
Kruskal–Wallis test	14.138		Kruskal–Wallis test	13.532	
*p*	0.003	(Chi-square approximation)	*p*	0.004	(Chi-square approximation)
Group	N	Mean rank of seed density in 10–20 cm layer	Group	N	Mean rank of seed density in 10–20 cm layer
<50% cover of wasteland	4	2.63	<50% cover of wasteland	4	4.50
>75% cover of wasteland	4	6.38	>75% cover of wasteland	4	4.50
<50% cover of cultivated land	4	10.50	<50% cover of cultivated land	4	11.50
>75% cover of cultivated land	4	14.50	>75% cover of cultivated land	4	13.50
Kruskal–Wallis test	13.988		Kruskal–Wallis test	13.492	
*p*	0.003	(Chi-square approximation)	*p*	0.004	(Chi-square approximation)

**Table 6 plants-13-02644-t006:** Kruskal–Wallis test results for diversity indices of weed communities within different habitats.

April			October		
Group	N	Mean rank of seed species richness (S)	Group	N	Mean rank of species richness (S)
<50% cover of wasteland	4	14.38	<50% cover of wasteland	4	12.63
>75% cover of wasteland	4	4.25	>75% cover of wasteland	4	7.5
<50% cover of cultivated land	4	8.38	<50% cover of cultivated land	4	8.75
>75% cover of cultivated land	4	7.00	>75% cover of cultivated land	4	5.13
Kruskal–Wallis test	10.017		Kruskal–Wallis test	5.407	
*p*	0.018	(Chi-square approximation)	*p*	0.144	(Chi-square approximation)
Group	N	Mean rank of Simpson index (D)	Group	N	Mean rank of Simpson index (D)
<50% cover of wasteland	4	14.5	<50% cover of wasteland	4	8.25
>75% cover of wasteland	4	6.5	>75% cover of wasteland	4	12.00
<50% cover of cultivated land	4	10.5	<50% cover of cultivated land	4	9.00
>75% cover of cultivated land	4	2.5	>75% cover of cultivated land	4	4.75
Kruskal–Wallis test	14.118		Kruskal–Wallis test	4.699	
*p*	0.03	(Chi-square approximation)	*p*	0.195	(Chi-square approximation)
Group	N	Mean rank of Shannon–Wiener index (H)	Group	N	Mean rank of Shannon–Wiener index (H)
<50% cover of wasteland	4	14.5	<50% cover of wasteland	4	10.75
>75% cover of wasteland	4	6.25	>75% cover of wasteland	4	11.00
<50% cover of cultivated land	4	10.5	<50% cover of cultivated land	4	7.50
>75% cover of cultivated land	4	2.75	>75% cover of cultivated land	4	4.75
Kruskal–Wallis test	13.787		Kruskal–Wallis test	4.654	
*p*	0.003	(Chi-square approximation)	*p*	0.199	(Chi-square approximation)
Group	N	Mean rank of Pielou index (J)	Group	N	Mean rank of Pielou index (J)
<50% cover of wasteland	4	14.5	<50% cover of wasteland	4	5.50
>75% cover of wasteland	4	6.5	>75% cover of wasteland	4	13.25
<50% cover of cultivated land	4	10.5	<50% cover of cultivated land	4	6.75
>75% cover of cultivated land	4	2.5	>75% cover of cultivated land	4	8.5
Kruskal–Wallis test		14.118	Kruskal–Wallis test	6.110	
*p*	0.003	(Chi-square approximation)	*p*	0.106	(Chi-square approximation)

## Data Availability

Data are contained within the article.

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
