# Peer review of "Effects of Acmella radicans Invasion on Soil Seed Bank Community Characteristics in Different Habitats"

_plants, 2024, doi:10.3390/plants13182644_

Round 1

Reviewer 1 Report

Comments and Suggestions for Authors

This manuscript describes the soil seed bank in two habitats (wasteland/cropland) colonized by the weed Acmella radicans. The authors used greenhouse germination tests as a method to assess soil bank parameters.

The study may be of local interest. However, it requires some improvements before publication.

Introduction:

The characterisation of the species (l. 74-80) should be completed with the species' habitat type and its concise morphological description. Are any methods applied against this plant where it is invasive?

Results:

In several places, the authors have included conclusions not supported by the results: l. 119- 124, l. 151-153.

Table 1 lacks complete species names (no authors) that have to be used, there are errors in names (e.g. Ranunculus japonicus - typos error here and  in subsequent tables). There is no need for the column with genus names (it follows from the species name), but there is no column for whether the species is native or alien or cultivated. It would be nice if for alien species the origin will be indicated. 

The order of species in the tables should be alphabetical - the same for the next tables.

Tables 4 and 5 - headings have to be corrected, as they are currently unreadable

Description of table 5 is incorrect – Distribution of Acmella radicans seeds in ….

Materials:

There is no description of soil types (l. 281), please complete.

General:

In the text, all Latin names should be in italics and in alphabetical order.

Language:

minor linguistic errors: l. 126, 129, l. 308, l. 159.

It should be used diversity in l. 165 (not bioviversity)

Comments on the Quality of English Language

Minor editing of English language required

Author Response

This manuscript describes the soil seed bank in two habitats (wasteland/cropland) colonized by the weed Acmella radicans. The authors used greenhouse germination tests as a method to assess soil bank parameters.

The study may be of local interest. However, it requires some improvements before publication.

Introduction:

The characterisation of the species (l. 74-80) should be completed with the species' habitat type and its concise morphological description. Are any methods applied against this plant where it is invasive?

Lines 80-83 We have provided some additional information about the species’ habitat type. As for the morphological description, we do not think it is necessary to include an elaborate description in the current manuscript as more is available in a previous paper if the reader is interested (Reference 15) and we do include the general morphological description of A. radicans as “an annual erect herb in the Asteraceae.” To date very little has been done to manage this plant in areas where it is invasive, so we were not able to include information on methods applied against this plant.

Results:

In several places, the authors have included conclusions not supported by the results: l. 119- 124, l. 151-153.

Deleted.

Table 1 lacks complete species names (no authors) that have to be used, there are errors in names (e.g. Ranunculus japonicus - typos error here and in subsequent tables). There is no need for the column with genus names (it follows from the species name), but there is no column for whether the species is native or alien or cultivated. It would be nice if for alien species the origin will be indicated. 

The order of species in the tables should be alphabetical - the same for the next tables.

Tables 4 and 5 - headings have to be corrected, as they are currently unreadable

Description of table 5 is incorrect – Distribution of Acmella radicans seeds in ….

We added the complete species names, changed the errors, deleted the column with genus names, added the origin, changed the order of species in the tables, and corrected the headings of tables.

Materials:

There is no description of soil types (l. 281), please complete.

We added some information in soil collection as follows: “The soils from the two locations were similar and identified as yellow earths in two invaded habitats. The soils nutrient levels were measured as follows: pH 4.63-4.67, organic matter 30.96-31.41 g/kg, total N 1.49-1.53 g/kg, total K 1.04-1.21 g/kg, and total P 13.57-13.99 g/kg.”

General:

In the text, all Latin names should be in italics and in alphabetical order.

Done.

Language:

minor linguistic errors: l. 126, 129, l. 308, l. 159.

Done.

It should be used diversity in l. 165 (not bioviversity)

Done.

Comments on the Quality of English Language  Minor editing of English language required

English has been improved by a native English speaker, David Clements, one of the co-authors of the manuscript.

Reviewer 2 Report

Comments and Suggestions for Authors

General opinion

There are many errors in the investigation methods. The categories established for the seed bank tests are too large (there are two categories in total: less than 50% plant cover and more than 75% cover). However, according to the authors, one plant can produce up to 14,300 seeds. The control is missing, i.e. a soil sample collected from an area with 0% coverage of the investigated invasive plant. The authors did not perform soil tests, so there is no information on how different the nutrient supply capacity of the two tested soils is. On what basis do the authors claim that the soils are similar? It can be assumed that there can be big differences between cultivated and uncultivated soil, especially in the topsoil. Knowing the most important soil parameters would also have been important for the germination experiment. Unfortunately, this test cannot be evaluated, because no control soil was used here either. It is not clear how natural the studied plain is. Has it been grazed a long time ago? A field that was once cultivated and left alone? This is an important aspect when examining the host capacity of an invasive species. Furthermore, the proportion of open, unvegetated spots is also important. The authors did not perform such tests. Due to the above problems, I see no point in further, substantial improvement of the work, so I do not recommend its publication.

Author Response

  1. Our study was focused on the soil seed banks of two different invaded habitats within different months, so the categories established (represent different invaded degree) for the seed bank tests are suitable. Moreover, many studies on soil seed banks examine one habitat type without control treatments. Given seed dormancy and seed movement by wind dispersion it is difficult to say that some plots are controls even if the plant is not present above-ground. Furthermore our study looked at natural fields rather than analyzing experiments with cultured plants, so it did not make sense to include controls.
  2. We examined the effects of different environmental factors on seed germination of radicansin previous work, and did not find differences in seed germination among soil types such maize soil, rice soil, wasteland soil and humus soil. Thus the impacts of soil parameters on soil seed germination of A. radicans do not appear to be very pronounced. Moreover, we did examine soil type, nutrient content and macrogenomic sequencing analysis of the two different invaded habitats reported on in the present study, but plan to report these results elsewhere. We did add some information in the materials as follows: “The soils from the two locations were similar and identified as yellow earths in two invaded habitats. The soils nutrient levels were measured as follows: pH 4.63-4.67, organic matter 30.96-31.41 g/kg, total N 1.49-1.53 g/kg, total K 1.04-1.21 g/kg, and total P 13.57-13.99 g/kg.”
  3. The wasteland was never grazed and/or cultivated since 2002. Thus local the spread in this habitat by radicansoccurred naturally and was not affected by human disturbance. This made for ideal conditions to compare of the wasteland and cultivated land. We agree that the host capacity of an invasive species was affected by many aspects such as habitats (open, unvegetated spots, roadsides, riverbank, etc.), human disturbance, climate conditions, and ecosystem type. However, we felt that the comparison of the two key habitats where A. radicans occurred in Yunnan provided valuable insights on the invasion biology of A radicans.

Reviewer 3 Report

Comments and Suggestions for Authors

The article is relevant and in line with the scope of the journal. It addresses important issues of seed bank structure. Without going into the merits of the article, I would like to make a few comments which, if taken into consideration by the authors, would further improve the quality of the article. 

1. Lines 111-127 and following throughout the text. Abbreviation of tribal names is unacceptable. In accordance with the requirements set out in the Nomenclature Code (https://www.iapt-taxon.org/nomen/pages/main/art_2.html), the abbreviation of a genus name is allowed (but not required!) only when the abbreviation does not cause ambiguity. The text now reads: 'A. conyzoides, A. radicans, B. pilosa, C. virgata, D. sanguinalis, Eleusine indica, [...]'.  The abbreviation of a genus name may only be used as long as no other genus name is mentioned. Now, it may be understood that "A conyzoides" is "Acmella conyzoides", but that is not true! The addition of a dozen or so more words will certainly not lengthen the text of the paper significantly, but it will make it easier to read and will not contravene the rules of nomenclature. 

2. In Table 1, the word herb is redundant in the Life form column because all the plants in the list are herbs. It would be sufficient to mention 'herb species' in the table caption.  

3.  "Bidens Pilosa L." spelling needs to be corrected in all tables and text. It should read Bidens pilosa L.

4. In the discussion, I suggest that attention should also be paid to the distribution of seed density and seed viability in other plants. I think it is worth discussing the differences and similarities between the seed density of herbaceous plants and some invasive shrubs (e.g. 10.3390/d14060488). Such comparisons are particularly useful when similar patterns emerge.

5. In my opinion, it is better to write 'sampling plots' rather than 'quadrats' in the methods when describing soil sampling (Line 307 and in the further text). 

6. How long was the soil stored before the start of the sampling experiment? Could the length of storage have had an effect on the germination of the seeds? (Line 314). 

7. It would be useful to mention exactly when the seed germination experiment was started and finished. This information is also important for other researchers wishing to plan similar studies (Line 327).

8. Line 429. The first name "Petr" of Pyšek should be deleted. (Pyšek P.).

Comments on the Quality of English Language

Minor editing of English language required

Author Response

The article is relevant and in line with the scope of the journal. It addresses important issues of seed bank structure. Without going into the merits of the article, I would like to make a few comments which, if taken into consideration by the authors, would further improve the quality of the article. 

1. Lines 111-127 and following throughout the text. Abbreviation of tribal names is unacceptable. In accordance with the requirements set out in the Nomenclature Code (https://www.iapt-taxon.org/nomen/pages/main/art_2.html), the abbreviation of a genus name is allowed (but not required!) only when the abbreviation does not cause ambiguity. The text now reads: 'A. conyzoides, A. radicans, B. pilosa, C. virgata, D. sanguinalis, Eleusine indica, [...]'.  The abbreviation of a genus name may only be used as long as no other genus name is mentioned. Now, it may be understood that "A conyzoides" is "Acmella conyzoides", but that is not true! The addition of a dozen or so more words will certainly not lengthen the text of the paper significantly, but it will make it easier to read and will not contravene the rules of nomenclature. 

We have carefully checked and revised all plant names by with full Latin names of all plants. 

2. In Table 1, the word herb is redundant in the Life form column because all the plants in the list are herbs. It would be sufficient to mention 'herb species' in the table caption. 

We kept the life form column because even though herbaceous the plant species differ in terms of being either annual or perennial.

3. "Bidens Pilosa L." spelling needs to be corrected in all tables and text. It shouldread Bidens pilosa L.

Done.

4. In the discussion, I suggest that attention should also be paid to the distribution of seed density and seed viability in other plants. I think it is worth discussing the differences and similarities between the seed density of herbaceous plants and some invasive shrubs (e.g. 10.3390/d14060488). Such comparisons are particularly useful when similar patterns emerge.

The distribution of seed density and seed viability would be useful to study but it was not part of our study so goes beyond its scope.

5. In my opinion, it is better to write 'sampling plots' rather than 'quadrats' in the methods when describing soil sampling (Line 307 and in the further text). 

We kept the terminology as quadrats.

6. How long was the soil stored before the start of the sampling experiment? Could the length of storage have had an effect on the germination of the seeds? (Line 314). 

The soil samples were collected in April and October 2022 and stored for one year before the sampling experiments. During another seed germination experiments, we found that the seed germination of A. radicans was mostly decided by water and light, not the storage length.

7. It would be useful to mention exactly when the seed germination experiment was started and finished. This information is also important for other researchers wishing to plan similar studies (Line 327).

Done. Germination tests were conducted in the greenhouse to estimate the seed bank composition and density from 5th June to 13th October, 2023.

8. Line 429. The first name "Petr" of Pyšek should be deleted. (Pyšek P.).

Done.

Round 2

Reviewer 1 Report

Comments and Suggestions for Authors

The authors have made a great effort to improve the text. In its present form it is better. However, the text should be read carefully and existing errors should be corrected e.g. Bidens Pilosa - tab 1,2,3, inconsistency in spelling of names e.g. once R.K. Jansen and once R. M. King or J. R. I.  Wood, once and H. Rob and once & J. and so on.

Comments on the Quality of English Language

The quality of English is ok.

Author Response

The authors have made a great effort to improve the text. In its present form it is better. However, the text should be read carefully and existing errors should be corrected e.g. Bidens Pilosa - tab 1,2,3, inconsistency in spelling of names e.g. once R.K. Jansen and once R. M. King or J. R. I.  Wood, once and H. Rob and once & J. and so on.

We have carefully checked and revised all plant names.

Reviewer 3 Report

Comments and Suggestions for Authors

The manuscript has undergone very few revisions since the first round of reviews. Most of the corrections are limited to language editing issues or corrections of technical errors identified in a previous review. It is regrettable that the significant comments made in the previous review have not been taken into account and that the responses are completely unjustified, I would even say bizarre.

1. In the previous review I clearly stated: "The abbreviation of a genus name may only be used as long as no other genus name is mentioned." This means that if another genus name is used, the genus name of Acmella must be written unabbreviated and then can be abbreviated again. After all, these are elementary rules of nomenclature that any specialist in the study of organisms must know.  

2. I did not understand the authors' reply that they did not study the density of seeds in the seed bank, and therefore they do not extend the discussion to a comparison of the density of the seed bank with the results of other studies. How then to understand the whole results section, which deals with seed density in the soil in different layers? I continue to consider that one of the major shortcomings of the discussion is that there is no more detailed comparison with the vertical distribution of seeds of other plants in the soil layers. 

3. In a previous review I wrote: 'In my opinion, it is better to write 'sampling plots' rather than 'quadrats' in the methods when describing soil sampling'. I continue to believe that the term is not appropriate. It describes the shape of the sampling plot but does not indicate that it is a sampling plot. There is a fundamental logical difference between the terms. 

4. The validity of the statistical methods is very much in doubt. The authors write that two-way ANOVA was used, but it is only appropriate when the data sets are normally distributed. From the results presented, I cannot believe that the data could be normally distributed and that parametric methods of analysis can be used. I believe that the non-parametric method of analysis and Dunn's post-hoc, which is insensitive to null values and does not produce false similarities or false differences, should have been used. Furthermore, there is no mention at all of what the distributions of the data were?

Comments on the Quality of English Language

Minor revisions required.

Author Response

1. In the previous review I clearly stated: "The abbreviation of a genus name may only be used as long as no other genus name is mentioned." This means that if another genus name is used, the genus name of Acmella must be written unabbreviated and then can be abbreviated again. After all, these are elementary rules of nomenclature that any specialist in the study of organisms must know.  

The genus and species names together comprise the scientific name of plants that the genus abbreviation does not affect their identification. Usually, when plant names appeared at first time it must be written unabbreviated and then should be abbreviated. The plant name at the beginning of a sentence should not be abbreviated. Thank you for your previous suggestion referring to the case when there are other plants with a Genus name beginning with “A” – this enabled us to avoid the ambiguity of some plant names, but we realized that we could still use A. radicans most of the time. We think the newly revised manuscript now uses abbreviations correctly.

2. I did not understand the authors' reply that they did not study the density of seeds in the seed bank, and therefore they do not extend the discussion to a comparison of the density of the seed bank with the results of other studies. How then to understand the whole results section, which deals with seed density in the soil in different layers? I continue to consider that one of the major shortcomings of the discussion is that there is no more detailed comparison with the vertical distribution of seeds of other plants in the soil layers. 

We understand the rationale behind the suggestion to compare total seed bank densities and the total weed seed distribution in soil layers with the results of other studies, but such a comparison was inappropriate for our study design and main objective to look at the impact of Acmella radicans on the seed bank community. The data encompassed multiple levels in the experimental design, including different species (28 plant species), two habitats (wasteland and cultivated land), different plant cover (<50% cover and >75% cover), two growth periods (April and October), and three soil layers (0-5, 5-10 and 10-20), so we did not include all possible comparisons in the manuscript. From our data analysis, we observed the seeds of all dominant species were focused on the 0-5 cm soil layer, so we utilized the total seeds of each species among 0-5 cm, 5-10 cm and 10-20 cm layers in the analysis. As a new invasive species, the relationship of this species and other plants was the highest priority in our analysis, and thus comparing the vertical distribution of seeds of other plants would require a different study design and rationale, potentially as a future study of soil seed banks under these circumstances.

3. my opinion, it is better to write 'sampling plots' rather than 'quadrats' in the methods when describing soil sampling'. I continue to believe that the term is not appropriate. It describes the shape of the sampling plot but does not indicate that it is a sampling plot. There is a fundamental logical difference between the terms. 

Agreed and changed.

4. The validity of the statistical methods is very much in doubt. The authors write that two-way ANOVA was used, but it is only appropriate when the data sets are normally distributed. From the results presented, I cannot believe that the data could be normally distributed and that parametric methods of analysis can be used. I believe that the non-parametric method of analysis and Dunn's post-hoc, which is insensitive to null values and does not produce false similarities or false differences, should have been used. Furthermore, there is no mention at all of what the distributions of the data were?

We did not write that the two-way ANOVA was used in the manuscript. We think our statistical method with one-way analysis of variance (ANOVA) was suitable for this study.

Comments on the Quality of English Language

Minor revisions required.

Some minor revisions in the use of English were completed.

Round 3

Reviewer 3 Report

Comments and Suggestions for Authors

When using either one-way or two-way ANOVA, the normality of the data is a necessary requirement. Your methodology is silent on the evaluation of the data, and I see from your results that your original data were not normally distributed, so that non-parametric methods of analysis had to be used (lines 332-334).  As a consequence, all the comparison results presented in Tables 2 to 5 are potentially erroneous, as they were obtained using a non-parametric method of analysis (one-way ANOVA). 

Although I disagree with some of the points I have made above but which the authors have rejected, these are the authors' views and they are free to make different judgements. However, the fundamental requirements of statistical methods cannot be ignored as they completely change the results and lead to erroneous conclusions.

Comments on the Quality of English Language

Minor revisions (what it means? [...] density/%; Header row of Table 5).

Author Response

Dear reviewer, thank you very much for your great suggestion on the data analysis. We tested the normality of the soil seed bank data and found the data were not normally distributed. Therefore, the nonparametric Kruskal–Wallis test was used for the identification of significant differences among groups at a 5% level using IBM SPSS22.0 software (Armonk, NY, USA). The revised manuscript includes tables of the Kruskal-Wallis results (see Tables 4-6 in the new version). For the plant density of each species we used the mean value that similarly many soil seed bank published papers did (i.e., Beata Feledyn-Szewczyk et al., Weed Flora and Soil Seed Bank Composition as Affected by Tillage System in Three-Year Crop Rotation. Agriculture 2020, 10: 186). We also changed the percentage data in tables 4 and 5 from the previous version into bar charts.

Comments on the Quality of English Language

Minor revisions (what it means? [...] density/%; Header row of Table 5).

Done.

Round 4

Reviewer 3 Report

Comments and Suggestions for Authors

I have no additional comments on the manuscript. 

Comments on the Quality of English Language

No additional comments.

Author Response

Dear reviewer, thank you very much for your great suggestion for improving the quality and readability of the manuscript.